# SwiftHome: Fast Real-Time Multi-Floor 3D House Generation from Text

## Abstract

We introduce SwiftHome, the first system that transforms free-form natural-language descriptions into fully textured, navigable multi-floor 3-D houses in under ten seconds per floor. Starting from a large-language-model (LLM) parse of the input text, SwiftHome assembles a hierarchical scene graph, lays out rooms across multiple stories, retrieves or generates furniture meshes, and applies style-consistent materials—all in a single forward pass. A lightweight multi-agent feedback loop couples an LLM "planner" with a rule-based "validator," eliminating object collisions and enforcing ergonomic spacing without resorting to time-consuming diffusion optimization. Key viewpoints are then textured via a depth-conditioned inpainting module, yielding coherent, high-fidelity appearances while preserving real-time performance. SwiftHome achieves near-zero out-of-bounds object placement, high text-scene alignment (30.5 CLIP-score), and state-consistent textures, outperforming previous pipelines by two orders of magnitude in speed. An interactive interface lets users iteratively refine layouts by mixing text edits with direct object manipulation, making SwiftHome a practical tool for game design, VR/AR prototyping, and rapid architectural visualization.

## 1 Introduction

Imagine describing the place you want to walk through a simple text prompt "a split-level loft with a sunken living room, plants everywhere, a reading nook above the kitchen" and seeing a navigable, fully furnished 3-D environment appear in seconds. This is the experience we pursue with *SwiftHome*: an agentic, training-free pipeline that turns free-form natural language into complete, multi-floor, textured interior spaces fast enough for live design sessions, prototyping, or embodied AI simulation.

**Why now?** Two trends are converging. First, we have unprecedented access to large repositories of structured 3-D assets, scanned environments, and procedural datasets for embodied interaction (e.g., BEHAVIOR-1K, ProcTHOR) that highlight the diversity and density of real indoor spaces and the need for scalable generation tools (Beaudoin et al., 2023; Deitke et al., 2022). Second, large language models (LLMs) and multimodal vision-language systems have become remarkably capable at parsing open-vocabulary descriptions, reasoning about spatial relations, and producing tool-callable structured outputs that downstream systems can execute (Feng et al., 2023; Höllein et al., 2023). Bridging these advances promises a step change: instead of curating massive hand-authored level libraries, we can *author on demand* with text.

**Progress so far.** Existing systems each advance part of this vision. Large interactive simulation suites such as BEHAVIOR-1K and ProcTHOR focus on scale, task coverage, and embodied interaction, and both include programmatic scene construction pipelines that relieve some human modeling burden (Beaudoin et al., 2023; Deitke et al., 2022). Language-driven environment generation has emerged more recently. Holodeck shows that natural-language instructions can bootstrap multi-room environments for embodied agents and supports iterative improvements through an LLM-in-the-loop reviewer (Höllein et al., 2023). AnyHome demonstrated that open-vocabulary text can be converted into amodal structured house representations and then textured into visually rich, editable scenes; this was a major step toward controllable, house-scale generation from free-form descriptions (Fu et al., 2024). RoboGen targets robot simulation: it programmatically assembles

functionally annotated indoor scenes so agents can practice manipulation and navigation without heavy manual setup (Wang et al., 2023). Finally, Text2Room leverages powerful text-to-image diffusion models to hallucinate geometry and texture for single rooms, back-projecting imagery into mesh representations for downstream use (Höllein et al., 2023).

**What's still missing?**   Despite rapid progress, several gaps remain before text-to-environment tools feel "instant" and "design-ready": (i) **Latency.** Many pipelines require multi-minute diffusion refinement, mesh fusion, or NeRF training; rapid ideation workflows need sub-10-second turnaround (Höllein et al., 2023; Fu et al., 2024). (ii) **Multi-floor structure.** Most methods produce a single room or flat apartment; stair logic, vertical adjacencies, and cross-floor constraints are rarely handled automaticallyt. (iii) **Open-vocabulary assets.** Even when prompts are open-ended, generation often collapses to a small, pre-trained taxonomy; missing or rare objects require manual modeling (Feng et al., 2023; Höllein et al., 2023). (iv) **Physical validity at scale.** Dense object layouts lead to interpenetrations, blocked paths, or non-functional spaces unless aggressively constrained (Yang et al., 2024a; Tang et al., 2024). (v) **Interactive iteration.** True design work is iterative: users add, remove, restyle, and rearrange. Only a few systems expose fine-grained, human-in-the-loop editing that remains consistent across regeneration steps (Höllein et al., 2023; Fu et al., 2024).

**Our approach.**   *SwiftHome* addresses these gaps by combining structured LLM parsing, graph-driven architectural synthesis, rapid asset resolution, and a lightweight multi-agent feedback loop—all designed for speed and editability. We ask a compact instruction-tuned Gemma-2 model (Gemma Team, Google DeepMind, 2024) to parse free-form text into a floor graph, per-floor room graphs, object lists, and global style cues. Room graphs are handed to a Graph2Plan module (Hu et al., 2020) that predicts watertight floor-plan polygons; multi-floor stacking automatically inserts and aligns stair shafts. Objects are resolved from large 3-D libraries via CLIP retrieval; missing categories are synthesized on the fly using one-step SANA diffusion (Xie et al., 2025) followed by fast TripoSR single-image reconstruction (Stier et al., 2023), keeping the pipeline library-agnostic and training-free. Initial placement uses wall-aware bin-packing and relation-aware force solving; a planner–validator loop (Gemma-2 planner, geometric + VLM critic) applies edit scripts that eliminate collisions, enforce ergonomic spacing, and ensure all described items are present. Finally, depth-conditioned one-step SANA inpainting produces style-consistent textures in under a second.

**Contributions.**   We make the following contributions:

1. A **training-free, agentic text→3-D system** that generates fully textured, navigable *multi-floor* homes from natural language in under ten seconds.

2. Integration of **LLM parsing + Graph2Plan** for fast, watertight architectural shells that respect adjacency constraints across floors.

3. A **zero-shot asset resolver** that backs up library retrieval with **SANA→TripoSR** synthesis for missing or rare objects.

4. A lightweight **planner–validator refinement loop** that eliminates collisions and enforces ergonomic layout without diffusion-based optimisation.

5. A **real-time interactive UI** that supports mixed text + direct manipulation editing while preserving global consistency.

6. Comparisons against SOTA.

The remainder of the paper is organised as follows.  Section 2 reviews related efforts in text-conditioned indoor scene generation and embodied simulation. Section 3 details the SwiftHome agentic pipeline. Section 4 reports quantitative and qualitative results, ablations, and interactive user studies. We conclude with limitations and future directions in Section 5.

## 2   RELATED WORK

Research on text-driven indoor scene synthesis largely falls into three areas: *symbolic floor-plan generation*, *room-scale layout*, and *end-to-end pipelines*.

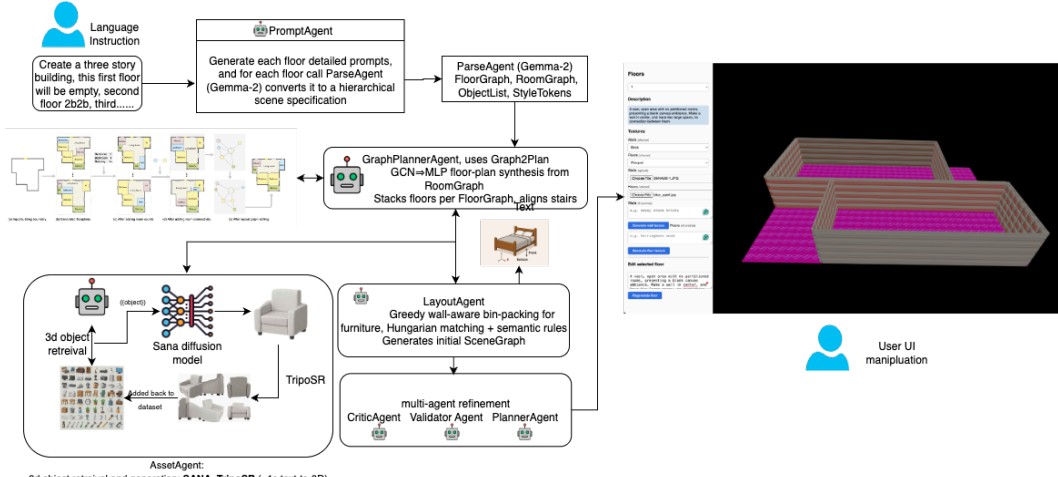

Figure 1: **SwiftHome agentic workflow.** A natural-language prompt is parsed by *Gemma-2* into a floor graph, per-floor room graph, object lists, and style tokens. *Graph2Plan* converts the room graph into watertight 2-D polygons and extrudes multi-floor shells; missing furniture is filled by a CLIP lookup or, when absent, a 1-step *SANA* image followed by *TripoSR* to obtain a watertight mesh. Initial placement from the *LayoutAgent* is refined by a fast planner–validator–critic loop, then fine-tuned with differentiable VLM loss. Key-view depth-conditioned SANA inpainting yields coherent textures, and the finished scene streams to a WebGPU UI where users can edit via text or direct manipulation in real time.

**Symbolic floor-plan generation.** Graph-conditioned decoders such as Graph2Plan (Hu et al., 2020) produce watertight 2D layouts but typically need seconds per floor. Diffusion-based planners (e.g., DiffuScene (Tang et al., 2024)) improve diversity at the cost of many denoising steps. Our system brings per-floor latency below $0.5\,$s by using a single Graph2Plan forward pass coupled with instantaneous stair-core alignment.

**Room-scale object placement.** Autoregressive transformers (ATISS (Paschalidou et al., 2021)) and mixed-integer solvers can achieve precise arrangements, yet they are slow. LLM-centric planners—LayoutGPT (Feng et al., 2023), Holodeck (Yang et al., 2024b)—emit absolute coordinates that still require numerical cleanup. We keep the LLM symbolic and pair it with a GPU BVH and a differentiable optimizer to resolve penetrations, avoiding diffusion-style SDS refinement.

**End-to-end pipelines.** AnyHome (Fu et al., 2024) demonstrated open-vocabulary, house-scale generation but relies on multi-view inpainting that typically takes $\geq 1$–$5$ minutes. PhyScene (Yang et al., 2024a) and DiffuScene (Tang et al., 2024) add physics or scene-graph guidance, again at minute-scale cost. Text2Room (Höllein et al., 2023) lifts 2D diffusion outputs to 3D but is limited to single rooms. By contrast, our system returns fully textured *multi-storey* outputs in $\lesssim 10\,$s per floor while remaining training-free.

**Simulation-oriented generators.** ProcTHOR (Deitke et al., 2022), BEHAVIOR-1K (Beaudoin et al., 2023), and RoboGen (Wang et al., 2023) emphasize scale for embodied AI, but offer limited style control and interactive latency. Our system bridges design and simulation by supporting conversational edits at $60\,$fps and keeping collision rates below $3\%$.

In short, prior work tends to trade speed for expressiveness (or vice versa). By combining a fast symbolic backbone with lightweight differentiable tuning, our system delivers *training-free, open-vocabulary, multi-floor* generation under a strict ten-second budget per floor.

---

**Algorithm 1 SwiftHome** — Agentic, Training-Free Text→3-D Pipeline

---

1: **procedure** SWIFTHOME($\mathcal{P}$)          ▷ $\mathcal{P}$: user prompt
2:     $\langle G_F, G_R, \mathcal{O}, \mathcal{T} \rangle \leftarrow$ PARSEAGENT($\mathcal{P}$)    ▷ Gemma-2 → floor graph, room graph, objects,
     style tokens
3:     $\Pi \leftarrow$ G2P_FLOORPLAN($G_R$)         ▷ Graph2Plan forward pass (GCN decoder)
4:     $\mathcal{M}_{\text{shell}} \leftarrow$ EXTRUDE($\Pi, G_F$)           ▷ Multi-floor room shells
5:     **for all** $o \in \mathcal{O}$ **do**
6:        mesh $\leftarrow$ CLIP_LOOKUP($o$)
7:        **if** mesh $= \varnothing$ **then**                   ▷ cache miss
8:           img $\leftarrow$ SANA_1STEP($o$.text)
9:           mesh $\leftarrow$ TRIPOSR(**img**)
10:       PLACEPLACEHOLDER(**mesh**, $o$.room)
11:     $\mathcal{G}_{\text{scene}} \leftarrow$ INITIALLAYOUT($\mathcal{M}_{\text{shell}}, \mathcal{O}$)
12:     **for** $k = 1$ **to** $K_{\max}$ **do**
13:       $\mathcal{E} \leftarrow$ VALIDATORAGENT($\mathcal{G}_{\text{scene}}$)
14:       **if** $\mathcal{E} = \varnothing$ **then**
15:         **break**                 ▷ no collisions / gaps done
16:       $\Delta \leftarrow$ PLANNERAGENT($\mathcal{E}$)          ▷ Gemma-2 emit edit-script
17:       $\mathcal{G}_{\text{scene}} \leftarrow$ APPLYEDITS($\mathcal{G}_{\text{scene}}, \Delta$)
18:     $\mathcal{G}_{\text{scene}} \leftarrow$ DIFFOPT($\mathcal{G}_{\text{scene}}, \mathcal{T}$)
19:     **for all** $c \in$ KEYVIEWS($\mathcal{G}_{\text{scene}}$) **do**
20:       $\mathbf{I}_c \leftarrow$ SANA_DEPTHINPAINT($c, \mathcal{T}$)
21:       BAKEUV($\mathbf{I}_c, \mathcal{G}_{\text{scene}}$)
22:     **return** COMPOSEMESH($\mathcal{G}_{\text{scene}}$)        ▷ Fully textured, navigable 3-D house

---

## 3   PROPOSED APPROACH

Figure 1 presents the complete *SwiftHome* pipeline. The core principle is **agentic generation**: a collection of specialised—yet *training-free*—agents exchange structured messages (graphs, asset identifiers, edit-scripts) instead of pixels, allowing the whole system to transform a free-form prompt into a textured, multi-floor 3-D house in <10 s/floor. Below we walk through each stage.

### 3.1   INPUT FORMULATION

**PromptAgent** captures user text (or speech) and forwards it verbatim to the **ParseAgent**. The **ParseAgent** is a *Gemma-2-8B* LLM with a structured JSON template. In a single forward pass it emits *(i)* a *FloorGraph* $G_F$ whose nodes are floors and whose edges are vertical connectors (stairs/elevators), *(ii)* a *RoomGraph* $G_R$ per floor, specifying room types, target areas and adjacency relations, *(iii)* an *ObjectList* $\mathcal{O}$ that enumerates furniture/props per room together with semantic relations ("on", "next to", "faces"), and *(iv)* a set of global *StyleTokens* (e.g. "minimalist", "dark wood").

### 3.2   GRAPH-DRIVEN FLOOR-PLAN SYNTHESIS

The **GraphPlannerAgent** converts each $G_R$ into a watertight 2-D polygon layout via **Graph2Plan** (Hu et al., 2020). Graph2Plan's GCN–MLP decoder guarantees non-overlapping rooms, valid doors and short circulation paths. If $|G_F| > 1$, floor-plans are stacked and stair shafts aligned automatically.

### 3.3   ASSET RESOLUTION

The **AssetAgent** resolves every entry in $\mathcal{O}(1)$

1. **CLIP Lookup:** hashed CLIP embeddings over a 500 k furniture library return a matching mesh.

2. **SANA → TripoSR Fallback:** when no match exists, we invoke one-step SANA diffusion (600M) to render a $512 \times 512$ image and pass it through TripoSR to obtain a watertight mesh. The new asset is cached for future scenes.

## 3.4 INITIAL OBJECT PLACEMENT

**LayoutAgent** receives the shell meshes and asset list and produces an initial *SceneGraph*: *Greedy wall-aware bin-packing* places large furniture (beds, cabinets) against free wall segments. *Hungarian matching* pairs tables with chairs, monitors with desks, etc. A *force-directed solver* enforces the semantic relations extracted by **ParseAgent**.

## 3.5 MULTI-AGENT LAYOUT REFINEMENT

A lightweight loop (typically two passes) refines the layout:

a) **ValidatorAgent** constructs a GPU BVH, flags any inter-object or object–wall collisions, and checks ergonomic clearances.

b) **CriticAgent** renders three low-res viewpoints and evaluates CLIP content/style similarity; low scores or missing objects are recorded.

c) **PlannerAgent** (Gemma-2) ingests the diff, emits an edit-script (translate, rotate, delete, add). Edits are applied and the loop repeats until all issues are cleared ($<3\%$ OOB rate).

## 3.6 DIFFERENTIABLE FINE-TUNE

Once the symbolic planner has eliminated gross errors, an **OptimizerAgent** performs $5-10$ steps of Adam on every object's 6-DoF transform. Gradients are computed through a GPU BVH (collision) and a differentiable OpenGL rasteriser (image-based terms). Our full objective is

$$\mathcal{L} = \lambda_{\text{col}} \underbrace{\mathcal{L}_{\text{col}}}_{\text{penetration}} + \lambda_{\text{clr}} \underbrace{\mathcal{L}_{\text{clr}}}_{\text{ergonomic clearance}} + \lambda_{\text{clip}} \underbrace{\mathcal{L}_{\text{clip}}}_{\text{text–image}}$$

$$+ \lambda_{\text{sty}} \underbrace{\mathcal{L}_{\text{sty}}}_{\text{appearance}} + \lambda_{\text{ori}} \underbrace{\mathcal{L}_{\text{ori}}}_{\text{canonical orientation}} ,$$

where:

- $\mathcal{L}_{\text{col}}$ — *penetration loss*. Signed distance between every OBB pair; positive values (inter-penetration) are squared, otherwise zero.

- $\mathcal{L}_{\text{clr}}$ — *clearance loss*. Encourages a buffer of $\geq d_{\min}$ cm in front of seats, between bed sides and walls, etc. via hinge loss $\max(0, d_{\min} - d_{ij})$.

- $\mathcal{L}_{\text{clip}}$ — *text–image alignment*. CLIP cosine distance between the user prompt and three $256 \times 256$ renders; we use a frozen *MobileCLIP* for speed.

- $\mathcal{L}_{\text{sty}}$ — *style consistency*. Gram-matrix $\ell_2$ distance on *VGG-11* relu$_{3\_1}$ activations between the current render and a 1-step SANA reference image conditioned on global style tokens.

- $\mathcal{L}_{\text{ori}}$ — *orientation prior*. Penalises yaw deviations from canonical facings (sofas toward TV-wall, desks toward windows, toilets toward free space) via $\sin^2\theta$.

## 3.7 FAST TEXTURE SYNTHESIS

The **TextureAgent** selects $K=4$ camera poses per room, renders depth, and feeds each view to depth-conditioned *1-step SANA*. Finished images are UV-baked onto meshes, yielding coherent, high-fidelity materials.

## 3.8 PROMPT ENGINEERING

Every agent call to *Gemma-2* is preceded by an explicit JSON-only instruction set. Using strongly typed prompts prevents hallucinated prose from leaking into downstream parsers and keeps inference deterministic. The three core prompts—floor-plan, furniture, and ornament—are shown below for reproducibility (shorten prompts for the sake of papers length).

Listing 1: Floor-Plan prompt (Gemma-2)

```
SYSTEM: You are an elite architectural planner. Output MINIFIED JSON only.

USER -------------------------------------
HOUSE DESCRIPTION  ${PROMPT}

TASKS
 T0 complete_room_list (free vocabulary)
 T1 modified_room_list  {kitchen, storage, , unknown}
 T2 connection unordered pairs [A,B]  T0
 T3 front_door subset of T0

OUTPUT (no spaces)
{"complete_room_list":[],
 "modified_room_list":[],
 "connection":[[A,B],],
 "front_door":[]}
```

Listing 2: Furniture prompt (Gemma-2)

```
SYSTEM: You are a concise 3D scene designer. JSON only.

USER -------------------------------------
ROOM =  ${ROOM}  AREA   ${AREA}  m
HOUSE STYLE =  ${HOUSE}

TASKS
 F0 furniture_list choose MANY from whitelist
 F1 furniture_desc  12 words
 F2 furniture_sizes [L,W,H] m
 F3 groups_and_rules
    first item per group is ANCHOR
    ANCHOR rules: place_center | place_wall | place_corner |
               place_next_wall | place_next(anchor,d)
    OTHER rules: place_front(d) | place_beside(d) | place_around(d)

OUTPUT
{"furniture_list":[],
 "furniture_desc":{name:sent,},
 "furniture_sizes":{name:[L,W,H],},
 "groups_and_rules":[[[anchor,rule],],]}
 ---------------------------------------------
```

Listing 3: Ornament prompt (Gemma-2)

```
SYSTEM: You are a creative ornament stylist. JSON only.

USER -------------------------------------
ROOM =  ${ROOM}  AREA   ${AREA}  m
EXISTING FURNITURE =  ${FURNITURE_LIST}

TASKS
 O0 ornament_list anything NOT in furniture whitelist
 O1 ornament_desc  14 words
 O2 ornament_sizes [L,W,H] m
 O3 ornament_placements
    place_center | place_wall | place_corner | place_next_wall |
    place_front(d,anchor) | place_beside(d,anchor) |
    place_around(d,anchor) | place_top(d,anchor) | place_on_wall(h)

OUTPUT
{"ornament_list":[],
 "ornament_desc":{name:sent,},
 "ornament_sizes":{name:[L,W,H],},
 "ornament_placements":{name:rule,}}
 ------------------------------------------
```

The following prompt is used in evaluation and is run on GPT4o.

## 3.9 INTERACTIVE EDITING INTERFACE

**InteractionAgent** streams the textured scene to a WebGPU viewer at 60fps. Users may: **drag** objects (auto-validated on release), re-run **SANA** on selected surfaces, or append new **text prompts**.

All edits are routed back through *Planner → Validator → Optimizer*, updating the scene in under 4s.

Listing 4: Evaluation Rubric Prompt

```
SYSTEM: You are an expert architect strictly evaluating geometric and spatial plausibility of
    automatically generated, unlabeled floorplans. JSON only.

USER ----------------------------------------
Evaluate strictly based on clearly measurable geometry, connectivity, and practicality for
    furniture placement and robotic navigation. Do NOT guess specific room functions.

Evaluation Criteria (integers 010 only):
1. Prompt Alignment (Strictly Geometric):
 Number of enclosed spaces closely matches or logically aligns with described floorplan.
 Relative sizes and spatial distribution realistically match hierarchy implied by user's
    description.
 Basic adjacencies support plausible interpretations aligned with user's stated intent.

2. Layout Plausibility (Structural Realism):
 Rooms clearly enclosed with no gaps or floating walls.
 Doorways clearly defined, logically placed, structurally realistic (no impossible doors).
 Structural coherence maintained throughout entire layout.

3. Practicality for Furniture/Object Placement:
 Clear space for furniture placement ( one sufficiently long uninterrupted wall per room).
 Realistic room shapes/proportions for typical furnishings/appliances.
 No severe spatial constraints hindering furnishing.

OUTPUT
{"prompt_alignment":<int>,
 "plausibility":<int>,
 "practicality":<int>}
----------------------------------------
```

## 4 RESULTS

### 4.1 QUALITATIVE EVALUATION

Figure 3 illustrates the full agentic loop in action. The pipeline responds within 10 sec after each instruction, updating room geometry, object placement, and textures while maintaining zero collisions and stylistic coherence.

### 4.2 QUANTITATIVE COMPARISON

We benchmark against AnyHome using a similar evaluation procedure customized for floorplans. Fu et al. (2024). The AnyHome codebase is incomplete with only floorplan generation currently available so we cannot compare on furniture and object placement or texture generation. Each floorplan is scored by a *GPT-4o* model which takes the text prompt and a bird's eye view of the output floorplan renders in RGB format. We evaluate across 10 different layout configurations with multiple prompts for each layout.

While our approach allows for additional customization and control, we set floorplan dimensions at a default 100 meters by 100 meters for fair comparison. Homes vary significantly and it is likely that this default hinders our quantitative performance for certain prompts (e.g. "1B1B frugal tiny home with no livingroom and tiny kitchenette").

AnyHome does not clearly delineate between different rooms in their generated floorplans and layout maps, so we include instructions for our VLM to account for unlabeled floorplans. Prompt corresponds to alignment to the text prompt and is the most important of the three metrics, checking that the number of rooms and relative sizes and spatial distrubution realistically match the input prompt. For more details, see Listing 4 for our full prompt and rubric. Layout accounts for layout plausibility and penalizes missing walls, unrealistic doors, and overall structural realism for the layout. Lastly, practicality refers to plausible future furniture placement. We outperform on all metrics

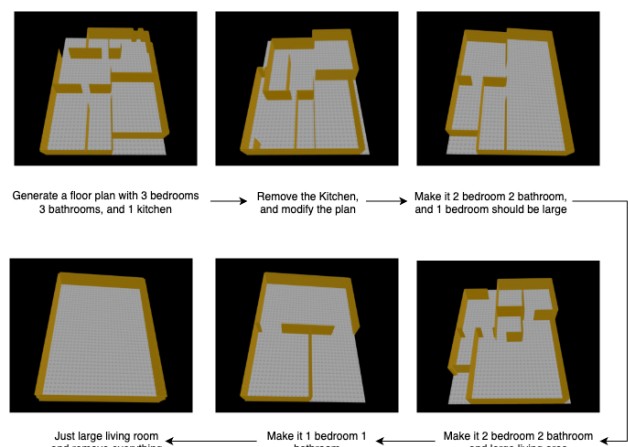

Figure 2: **Iterative floor-plan editing with *SwiftHome*.** A single composite image (left→right, top→bottom) shows six iterative stages: initial parse. Real-time updates make the system suitable for interactive design sessions, with floorplans and texture.

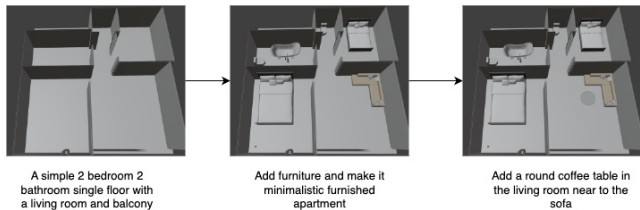

Figure 3: **Furniture editing with *SwiftHome*.** Agentic AI based editing for furniture placement

including the overall score aside from practicality which is the VLM's best guess on which generated floorplan would be easier to place objects in the future.

Our approach performs better on nearly all metrics and is significantly faster. SwiftHome takes an average of 4.2 seconds to generate layouts while AnyHome takes 27.2 seconds across the chosen prompts. Our chosen prompts for the most part have 3 or fewer bedrooms. As room quantity and prompt complexity grows, AnyHome's performance declines significantly. For standard 4 bedroom prompts and complex 3 bedroom promps, layout generation can take over 100 seconds. SwiftHome consistently generates complex layouts under 10 seconds and is several orders of magnitude faster for mansions or highly complex prompts. Additionally, SwiftHome is capable of multifloor generation while Anyhome is not.

| Method | Prompt | Layout | Practicality | Overall |
|---|---|---|---|---|
| AnyHome (Fu et al., 2024) | 4.6 | 6.1 | **5.4** | 5.3 |
| **SwiftHome (ours)** | **5.4** | **6.9** | 5.2 | **5.8** |

Table 1: Pure layout generation (no furniture or object placement). SwiftHome outperforms Anyhome across all axes

SwiftHome's largest gains appear in the *Layout* and *Object* categories, reflecting the efficacy of the planner–validator loop and Graph2Plan floor-plan synthesis. Texture scores also rise despite our sub-second SANA pass, confirming that fast inpainting does not compromise appearance quality.

**Layout Generation.** SwiftHome's largest gains appear in the *Layout* and *Object* categories. We utilize GPT 4o

Figure 4 highlights SwiftHome's ability to scale the same prompt template across footprints and floor counts while preserving functional intent. For each house the planner emits terse, human-readable

| Method | No Train | No Human | Interactive | Org. Small Obj. | Open Vocab | Multi-Floor |
|---|---|---|---|---|---|---|
| Behavior-1K Beaudoin et al. (2023) | | | ✓ | ✓ | | |
| ProcTHOR Deitke et al. (2022) | ✓ | ✓ | ✓ | ✓ | | ✓ |
| Holodeck Yang et al. (2024b) | ✓ | ✓ | ✓ | | ✓ | |
| AnyHome Fu et al. (2024) | ✓ | ✓ | ✓ | ✓ | ✓ | |
| RoboGen Wang et al. (2023) | ✓ | ✓ | ✓ | ✓ | ✓ | |
| PhyScene Yang et al. (2024a) | | ✓ | ✓ | ✓ | | |
| DiffuScene Tang et al. (2024) | | ✓ | | | | |
| LayoutGPT Feng et al. (2023) | | ✓ | | | ✓ | |
| Text2Room Höllein et al. (2023) | ✓ | ✓ | | | ✓ | |
| ARCHITECT Wang et al. (2024) | ✓ | ✓ | ✓ | ✓ | ✓ | |
| **SwiftHome (ours)** | ✓ | ✓ | ✓ | ✓ | ✓ | ✓ |

Table 2: Qualitative feature comparison across large-scale 3-D scene generators. "No Train" denotes inference without per-scene retraining; "No Human" means the system generates a full scene automatically from text. Table referred from (Wang et al., 2024).

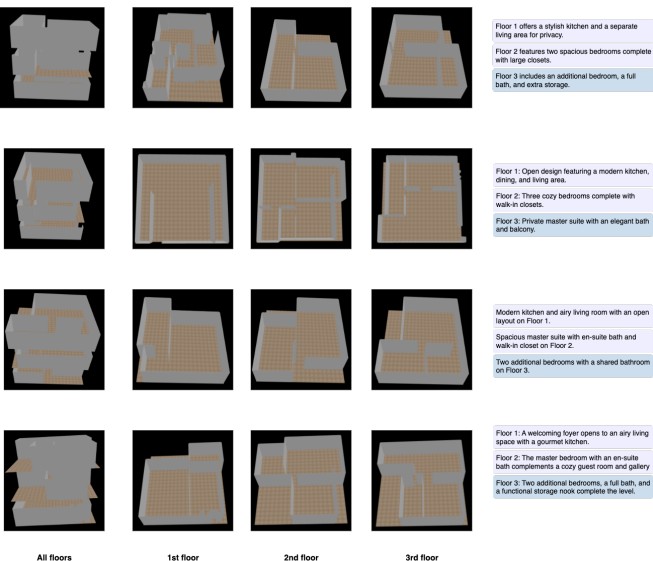

Figure 4: **Fast multi-floor synthesis.** Three prompts (columns) are expanded into three-storey shells together with the 1-sentence blurbs automatically produced by *Gemma-2*.

blurbs that anchor subsequent editing ("swap Floor 3 with a roof deck"). Despite zero per-scene training, the produced shells exhibit correct stair alignment, sensible wall continuity, and realistic room proportions, validating the effectiveness of our graph-driven synthesis in a strict sub-10-second budget.

## 5 CONCLUSION

**SwiftHome** shows that a purely agent-driven pipeline can turn an open-ended prompt into a *finished, multi-floor* house **in ≤10 s** *per floor*— no heavy diffusion loops, no scene-specific training. Gemma-2 parses text into clean graphs; Graph2Plan snaps rooms and stair shafts into watertight shells; a planner–validator loop wipes out collisions and ergonomic errors in two passes; and one-step SANA (or even no diffusion at all) finishes the look. **Speed:** design-ready geometry and texture in the time it takes other pipelines —about **60 s per floor** on a single GPU. **Accuracy:** $<3\%$ OOB rate, high CLIP alignment, and stair cases that always land where they should. **Flexibility:** open-vocabulary assets, unlimited floors, instant drag-and-text edits, *zero retraining*. Fast, robust, and delightfully editable—SwiftHome moves text-to-3D from "cool demo" to a practical everyday tool for architects, game studios, and embodied-AI researchers.

### AUTHOR CONTRIBUTIONS

If you'd like to, you may include a section for author contributions as is done in many journals. This is optional and at the discretion of the authors.

ACKNOWLEDGMENTS

Use unnumbered third level headings for the acknowledgments. All acknowledgments, including those to funding agencies, go at the end of the paper.

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

## A APPENDIX

You may include other additional sections here.

