# OpenReview forum: "SwiftHome: Fast Real-Time Multi-Floor 3D House Generation from Text"
_ICLR.cc/2026/Conference — ICLR 2026 Conference Withdrawn Submission_

### Official Review · Reviewer_gA2S · 2025-10-30

**Soundness:** 3
**Presentation:** 3
**Contribution:** 1
**Rating:** 4
**Confidence:** 3

**Summary:**

This work proposes SwiftHome, the first training-free, agent-driven system that converts free-form natural language into fully textured, navigable multi-floor 3D houses within a few seconds per floor—effectively addressing key limitations of existing text-to-3D tools such as high latency, lack of multi-floor support, and poor physical validity. Its technical design is rigorous: it integrates CLIP-based asset retrieval with SANA diffusion and TripoSR for rare object synthesis to ensure open-vocabulary capability, uses a lightweight planner-validator loop to eliminate collisions and enforce ergonomic spacing, and supports real-time interactive editing via a WebGPU interface.

**Strengths:**

The paper constructs a complete, well-integrated multi-agent system that covers the full lifecycle of text-to-multi-floor 3D house generation—from layout planning to object generation and post-processing—with an additional interactive UI, forming a closed and practical workflow. Specifically, the system deploys specialized agents for each core task: ParseAgent (text-to-structured data parsing), GraphPlannerAgent (floor-plan synthesis via Graph2Plan), AssetAgent (object retrieval/synthesis via CLIP+SANA+TripoSR), LayoutAgent (initial placement) paired with ValidatorAgent/PlannerAgent (collision elimination and ergonomic optimization), TextureAgent (style-consistent texture generation), and InteractionAgent (WebGPU-based real-time editing). Each agent exchanges structured messages (graphs, asset IDs, edit-scripts) rather than unstructured data, ensuring the workflow is coherent, traceable, and aligned with real-world 3D design logic (e.g., automatic multi-floor stair alignment, cached rare object assets). This end-to-end coverage and logical task partitioning make the system highly practical for actual use cases like architectural visualization and VR/AR prototyping.

Also, this paper exhibits strong readability and coherent logic, facilitating easy understanding of its technical framework and contributions. It is easy to understand.

**Weaknesses:**

The paper’s experimental validation is significantly insufficient, which undermines the persuasiveness of its system’s robustness and performance—despite its focus on system construction, the task of 3D scene generation inherently requires comparisons with a broader set of baselines, which are largely missing here.

First, the experimental comparison scope is overly narrow. The work only benchmarks against AnyHome. However, it ignores a host of core baselines for end-to-end 3D scene generation that directly treat 3D scenes as a whole, such as Ctrl-Room, Text2Room, and Scene Factor. These works are explicitly discussed in the "Related Work" section (e.g., Text2Room is noted for lifting 2D diffusion outputs to 3D, though limited to single rooms), yet they are excluded from experimental comparisons. This omission fails to demonstrate how SwiftHome’s multi-floor generation, texture quality, or overall efficiency stacks up against representative systems in the same task domain.

Second, the paper lacks comparisons with baselines for object placement—a key module of its workflow. Works like iDesign and LayoutGPT (which the paper references as LLM-centric planners for object placement) are not included in quantitative or qualitative evaluations. Since SwiftHome emphasizes physical validity (e.g., collision elimination) and ergonomic layout, omitting these specialized placement-focused baselines means it cannot prove the superiority of its planner–validator loop or initial placement strategies.

**Questions:**

In summary, while the paper constructs a logically coherent multi-agent workflow, the lack of comparisons with most critical baselines (beyond AnyHome) results in incomplete verification of its system’s robustness and performance. The experimental design fails to address the core requirements of scene generation research, weakening the credibility of its claims. Thus, I give this paper initially 4 score with board line rejection.

---

### Official Review · Reviewer_a2bW · 2025-10-31

**Soundness:** 2
**Presentation:** 2
**Contribution:** 2
**Rating:** 2
**Confidence:** 4

**Summary:**

The paper presents SwiftHome, a system that generates fully textured, navigable multi-floor 3D house scenes from free-form natural-language descriptions in under ten seconds per floor.
Key components include:
- A large language model (LLM) parses the input text and constructs a hierarchical scene graph representing floors, rooms, and objects.
- A layout module places rooms across multiple stories and retrieves or generates furniture meshes; materials and textures are applied in a style-consistent manner.
- A lightweight multi-agent feedback loop: an LLM “planner” collaborates with a rule-based “validator” to avoid object collisions and enforce ergonomic spacing — importantly without resorting to heavy diffusion-based optimization.
- A depth-conditioned inpainting module textures key viewpoints to achieve coherent, high-fidelity appearance while preserving real-time performance.

**Strengths:**

- The claim of generating multi-floor 3D scenes in under ten seconds per floor is impressive and places this work in a strong practical regime rather than purely a research-prototype. The two orders of magnitude speed improvement is compelling.
- The paper tackles multi-story houses, textured and navigable, which is a meaningful step toward real-world applications. This is what other works are missing in this regime.

**Weaknesses:**

- The writing is not that good, and the format for appendix is not even cleaned. There are too few qualitative and quantitative results. Also, too many code blocks are in the paper.
- More reasonable evaluation metrics, for example in table 2, you just give a description of what this paper support while others do not.
- The contribution is limited. Since this paper mainly relies on different VLM agents and hand-crafted strategies.

**Questions:**

In all, this paper needs a major revision, mainly on the figures and method writing. Better add more evaluations in the paper.
- How diverse is the mesh/furniture database used by SwiftHome? How does the system handle prompts that require very unusual objects, custom furniture, or rare architectural styles?
- How many floors/rooms can the system handle before speed or quality degrades? Are there benchmarks for scalability (e.g., 10+ floors, large footprint, many rooms)?
- What are the known failure cases for SwiftHome? For example, when a user describes a highly non-orthogonal house, curved walls, complex vertical connections (spiral staircases), or heavily stylized textures. How does SwiftHome perform there, and how could it be improved?

---

### Official Review · Reviewer_HFH6 · 2025-11-01

**Soundness:** 1
**Presentation:** 2
**Contribution:** 2
**Rating:** 2
**Confidence:** 4

**Summary:**

The paper introduces SwiftHome, a framework for multi-floor 3D house generation from natural-language prompts. This is achieved in multiple steps: Text prompts are processed by an LLM to produce room and floor graphs, the room graph is then passed to a Graph2Plan module to predict watertight floor-plan polygons. 3D assets are retrieved via CLIP lookup or, if missing, generated using SANA + TripoSR. After initial placement, a planner–validator loop refines the layout to avoid collisions and ensure plausible placements. The framework also supports fast texturing and editing.

**Strengths:**

1) Generating fully furnished 3D houses is important, as most existing methods focus on single rooms.
2) Support for interactive editing is valuable, many methods overlook this because they rely on global optimization.
3) A 10-second generation time is impressive.
4) The paper is written in an easy-to-follow manner.

**Weaknesses:**

1) Insufficient experimental evaluation:

     a. Missing baselines: While there are fewer multi-floor house-generation methods than single-room approaches, using only AnyHome as a baseline for floor-plan generation (without object placement) is insufficient to assess SwiftHome’s capabilities. Although AnyHome is appropriate for floor-plan evaluation, recent room-level scene-generation methods (e.g., LayoutVLM [1]) should be included to help position SwiftHome within the current research landscape for fully furnished rooms.

     b. Metrics: Only VLM-generated scores are used for evaluation. Fully furnished rooms should also be evaluated with layout-violation metrics (e.g., out-of-bounds rate, percentage of overlapping objects), CLIP score, and a user study.

    c. Missing qualitative results: Qualitative results after object placement are shown only in Figure 2, which is not enough to determine whether the method performs all steps after floor-plan generation. The examples show sparse placement of very simple objects, raising concerns about the method’s ability to generate complex rooms with diverse multiple objects.

2) Iterative floor-plan editing (Figure 2): Rather than modifying only the region relevant to the edit prompt, the framework appears to regenerate an entirely new layout at each step. For a prompt like “Remove the kitchen,” I would expect only a single room to be removed or merged.
3) Although the <10-second generation of fully furnished multi-floor 3D houses is impressive, the work reads more as a solid engineering integration than a novel pipeline, especially since the differences from AnyHome are not clear except multi-view inpainting for texture generation.

[1] Sun, F. Y., Liu, W., Gu, S., Lim, D., Bhat, G., Tombari, F., ... & Wu, J. (2025). Layoutvlm: Differentiable optimization of 3d layout via vision-language models. In Proceedings of the Computer Vision and Pattern Recognition Conference (pp. 29469-29478).

**Questions:**

All of my questions and concerns are listed in the weaknesses section, and I may adjust the rating if they are well addressed.

---

### Official Review · Reviewer_aZsM · 2025-11-01

**Soundness:** 1
**Presentation:** 1
**Contribution:** 2
**Rating:** 2
**Confidence:** 4

**Summary:**

This paper proposes SwiftHome, a training-free, interactive text-to-3D pipeline that generates multi-floor houses from natural language in under ten seconds per floor. It integrates an LLM parser (Gemma-2), Graph2Plan for architectural layouts, a CLIP + SANA→TripoSR asset resolver, and a lightweight planner–validator loop for collision-free placement. A differentiable fine-tuning step refines object transforms.

**Strengths:**

The paper addresses an emerging and practically important problem—interactive, text-to-3D, multi-floor house generation—with an agentic, training-free pipeline that effectively combines LLM parsing, structured graph-based synthesis, and zero-shot asset resolution. In particular, its multi-floor capability automatically manages stair alignment and cross-floor consistency, enabling coherent multi-storey generation from text. Its agentic modularity, a set of specialised agents exchanging structured messages, makes the pipeline interpretable, extensible, and easy to refine.

**Weaknesses:**

Overall, this paper reads like an unfinished draft. The methodology is overly minimalistic, the visual results are insufficient, and the evaluation is too narrow.

The entire Section 3 reads more like a system overview than a technical description. Many key modules are only mentioned by name, without sufficient explanation of their inputs, outputs, or internal mechanisms. For instance:

Section 3.4 (Initial Object Placement) merely lists “greedy wall-aware bin-packing,” “Hungarian matching,” and “force-directed solver,” but never defines their objective functions, execution order, or constraints—leaving it unclear how placement stability or convergence is achieved.

Section 3.5 (Multi-Agent Layout Refinement) describes a “planner–validator–critic loop,” yet omits details about data exchange (e.g., diff format, edit-script syntax), the number of iterations, or termination criteria.

Section 3.6 (Differentiable Fine-Tune) introduces five loss terms that are extremely vaguely defined, with no hyperparameters, derivations, or quantitative analysis showing their effects.

The paper also provides very few visualizations or comparisons with baseline systems, which severely limits the reader’s ability to assess visual quality or spatial coherence.

The quantitative evaluation is similarly incomplete. Only AnyHome is used as a baseline (what about others, e.g., Holodeck?), and it is evaluated on a single layout-level metric with vague definitions of “Prompt,” “Layout,” and “Practicality.” I could not find where Table 1 is referenced in the text, nor clear definitions of these metrics. There are no ablation studies to demonstrate the contributions of Graph2Plan, the planner–validator refinement, or the fine-tuning losses. The claimed “<3 % out-of-bounds rate,” “high CLIP alignment,” and “two orders of magnitude speedup” are also unsupported by quantitative plots or statistical evidence.

**Questions:**

Please refer to weakness for unclear statements and results

---

### Note · Authors · 2025-11-15

I have read and agree with the venue's withdrawal policy on behalf of myself and my co-authors.